# Identification of Novel Micropeptides Derived from Hepatocellular Carcinoma-Specific Long Noncoding RNA

**DOI:** 10.3390/ijms23010058

**Published:** 2021-12-21

**Authors:** Mareike Polenkowski, Sebastian Burbano de Lara, Aldrige Bernardus Allister, Thi Nhu Quynh Nguyen, Teruko Tamura, Doan Duy Hai Tran

**Affiliations:** 1Institut fuer Zellbiochemie, OE4310, Medizinische Hochschule Hannover, Carl-Neuberg-Str. 1, 30623 Hannover, Germany; Polenkowski.Mareike@mh-hannover.de (M.P.); sebastian.burbanodelaracarrillo@dkfz-heidelberg.de (S.B.d.L.); Allister.Bernardus@mh-hannover.de (A.B.A.); Nguyen.quynh@t-online.de (T.N.Q.N.); 2Systems Biology of Signal Transduction B200, German Cancer Research Center (DKFZ), Im Neuenheimer Feld 280, 69120 Heidelberg, Germany

**Keywords:** HCC-specific small functional protein, NONHSAT013026.2/Linc013026-68AA, fine tuner of cancer formation, dark proteome, hepatocellular carcinoma

## Abstract

Identification of cancer-specific target molecules and biomarkers may be useful in the development of novel treatment and immunotherapeutic strategies. We have recently demonstrated that the expression of long noncoding (lnc) RNAs can be cancer-type specific due to abnormal chromatin remodeling and alternative splicing. Furthermore, we identified and determined that the functional small protein C20orf204-189AA encoded by long intergenic noncoding RNA Linc00176 that is expressed predominantly in hepatocellular carcinoma (HCC), enhances transcription of ribosomal RNAs and supports growth of HCC. In this study we combined RNA-sequencing and polysome profiling to identify novel micropeptides that originate from HCC-specific lncRNAs. We identified nine lncRNAs that are expressed exclusively in HCC cells but not in the liver or other normal tissues. Here, DNase-sequencing data revealed that the altered chromatin structure plays a key role in the HCC-specific expression of lncRNAs. Three out of nine HCC-specific lncRNAs contain at least one open reading frame (ORF) longer than 50 amino acid (aa) and enriched in the polysome fraction, suggesting that they are translated. We generated a peptide specific antibody to characterize one candidate, NONHSAT013026.2/Linc013026. We show that Linc013026 encodes a 68 amino acid micropeptide that is mainly localized at the perinuclear region. Linc013026-68AA is expressed in a subset of HCC cells and plays a role in cell proliferation, suggesting that Linc013026-68AA may be used as a HCC-specific target molecule. Our finding also sheds light on the role of the previously ignored ’dark proteome’, that originates from noncoding regions in the maintenance of cancer.

## 1. Introduction

Hepatocellular carcinoma (HCC) is one of the most prevalent tumor types worldwide [1]; however, current treatment options are limited, and precise and effective medical strategies for therapy do not exist [2]. HCC typically occurs on a background of chronic liver disease, with risk factors including viral or autoimmune hepatitis, chronic alcohol abuse, and nonalcoholic fatty liver disease [3]. These risk factors trigger aberrant liver regeneration, which initiates the formation of HCC. However, the underlying molecular mechanism is still largely unknown. It has been recently shown by exome sequencing of HCC that 161 putative driver genes are associated with 11 recurrently altered pathways in HCC development, suggesting that many signaling pathways are altered to a modest degree, and act together [4,5,6]. Notably, 28% of altered gene products are involved in a chromatin-remodeling complex, suggesting that HCC expresses unique genes that are not expressed in normal hepatocytes. In this context, we have previously shown that a subset of lncRNAs that are predominantly expressed in HCC plays a role as fine tuners in cancer formation and/or maintenance [7,8]. Thus, a potential strategy for cancer therapy may be to target multiple cancer type-specific fine tuners including noncoding RNA.

Traditional annotation of protein-encoding genes relied on assumptions, such as one open reading frame (ORF) encodes one protein and minimal lengths for translated proteins [9]. However, recent data from our laboratory and from others have revealed that RNAs previously considered noncoding, such as long noncoding RNAs (lncRNAs) and circular RNAs are translated into functional small proteins [10,11,12,13], suggesting that the proteome is more complex than previously anticipated.

In the present study we utilized RNA sequencing (RNA-seq) and polysome profiling to identify novel micropeptides that originate from HCC-specific lncRNAs. We identified two HCC-specific lncRNAs that are translated into small ORFs. Applying a peptide specific antibody we characterized one lncRNA candidate, NONHSAT013026.2/Linc013026-68AA. Linc013026-68AA is translated into a 68 amino acid micropeptide that is mainly localized at the perinuclear region. Notably, Linc013026-68AA is predominantly expressed in moderately- but not well-differentiated HCC cells and plays a role in cell proliferation, suggesting that Linc013026-68AA may be used as a HCC-specific target molecule. Our data also uncover the important role of previously ignored small ORFs originating from noncoding regions in the maintenance of cancer.

## 2. Results

### 2.1. Identification of Hepatocellular Carcinoma-Specific lncRNAs

To identify lncRNAs that are expressed in HCC cells but not in normal hepatocytes, we used publicly available RNA-sequencing (RNA-seq) data generated by the ENCODE Consortium [14] to extract the expression level of lncRNAs. Firstly, we mapped RNA-seq data from normal liver (ENCFF184YUO) and from the HCC cell line HepG2 (ENCFF337WTM) to long intergenic non-coding RNAs (lincRNA) annotated by NONCODE v5.0, an integrated knowledge database of non-coding RNAs [15] (Figure 1A). We limited our study to lincRNAs, because RNA-seq based on second generation sequencing limits the accurate allocation of reads if lncRNAs overlap with coding genes. RNA-seq datasets were normalized using cuffnorm [16]. A total of 906 lincRNAs that expressed only in HepG2 cells but not in the liver were selected. To identify HCC-specific lincRNAs we further examined the expression of our lincRNA candidates in normal tissues including adipose, adrenal, brain, breast, colon, foreskin, heart, kidney, lung, ovary, placenta, prostate, skeletal muscle, testis and thyroid tissues and leukocytes using RNA-seq data generated by the Human Body Map 2.0 [17]. Twelve out of 906 lincRNAs were expressed exclusively in HepG2 cells (Figure 1A, Appendix A). We then confirmed the expression of these lincRNAs in HepG2 cells using our previously published RNA-seq data [7]. The expression of nine out of twelve HCC-specific lincRNAs was confirmed (Figure 1B, HepG2 (GSE115139)).

We next asked why these lincRNAs are expressed exclusively in HepG2 cells but not in the liver. We have previously demonstrated that altered chromatin structure in cancer results in the cancer-specific expression of a subset of genes [7,8]. Thus, we examined the chromatin structure at the putative promotor region of nine HCC-specific lincRNAs using DNase-sequencing data (DNase-seq) generated by the ENCODE Consortium [18]. DNase-seq data (Figure 1B, DNase-seq) obtained from human hepatocyte (ENCFF851CVH) and HepG2 (ENCFF474LSZ) revealed that HepG2 contains DNase I hypersensitive sites at the proximal promoter region of seven out of nine lincRNAs (except for NONHSAT204527.1 and NONHSAT223630.1 (Figure 1C)), while normal human liver does not contain these sites at these positions (Figure 1B, blue arrow), suggesting that the chromatin structure in this region is remodeled in HCC cells. To examine whether these open chromatin regions are associated with cis-regulatory elements, we utilized candidate cis-Regulatory Elements (cCREs) database generated by ENCODE consortium which contain 1,063,878 human cCREs [19]. Notably, putative promoter regions of five out of these seven lncRNA candidates contains at least one cCRE (Figure 1B, ENCODE cCRE, blue mark). In addition, ChIP-seq of H3K4 trimethylation and H3K27 acetylation, chromatin marks of active transcription revealed that these putative promoter regions of seven lncRNA candidates are transcriptionally active in HepG2 cells (Appendix A, H3K4me3 and H3K27Ac). These data suggest that transcription may be initiated from these regions. Thus, we utilized the cap analysis of gene expression (CAGE) data in HepG2 cells that mapped the transcription start sites (Figure 1B, CAGE). In agreement with RNA-seq data, the cap site is located at the open promoter regions determined by DNase-seq data (Figure 1B, Transcription Start Site (TSS-black arrow)), suggesting that altered chromatin structure plays a key role in the HCC-specific expression of lincRNAs.

In addition to chromatin structure, tissue-specific transcription factors (TFs) are well known to activate tissue-specific expression program [20]. Thus, we next examined which transcription factor potentially activates the transcription of HCC-specific lncRNA genes by utilizing ChIP-seq datasets of 340 factors generated by ENCODE consortium in HepG2 cells. All HCC-specific lncRNA genes are potentially activated by three to seven TFs (Appendix A). Notably, these TFs are expressed in both normal liver and primary HCC (Appendix A), suggesting that open chromatin structure at the promoter region rather than a transcription factor may play a key role in the HCC-specific expression of lncRNAs.

### 2.2. Identification of Micropeptide Candidates Derived from HCC-Specific lncRNAs

Six out of nine lincRNAs contain at least one open reading frame (ORF) that is longer than 50 amino acids (AA) (Table 1). To be translated into micropeptides, lincRNAs have to be exported to the cytoplasm. Thus, we examined the mRNA export of these 6 lincRNA candidates using nuclear- and cytoplasmic RNA-seq generated by the ENCODE Consortium. Except for NONHSAT142412.2 the other five lincRNAs were clearly detected in cytoplasmic RNA-seq (Figure 2A), suggesting that they are exported to the cytoplasm. We also confirmed the mRNA export using RT-PCR (Figure 2B). To examine whether these five lincRNA candidates are endogenously translated in HepG2 cells, we isolated the polysome fraction of HepG2 cells using sucrose gradient centrifugation [11] and performed qRT-PCR and RT-PCR for five lincRNAs. Actin mRNA was used as a positive control. Three out of five lincRNA candidates were detected in translated fractions of HepG2 cells (Figure 2C and Appendix A, NONHSAT013026.2, NONHSAT168790.1 and NONHSAT250607.1), suggesting that they are translated.

### 2.3. NONHSAT013026.2/Linc013026-68AA Is Translated into a 68 Amino Acid Long Micropeptide

Among three lincRNA candidates that were translated, NONHSAT013026.2 had the highest degree of enrichment in a translated fraction, thus we further focused on the characterization of this lincRNA which we renamed Linc013026. Linc013026 potentially encodes two ORFs of 52AA and 68AA. Since ORF-68AA has a Kozak consensus sequence, we further focused on this ORF. First, we examined whether Linc013026-68AA is translated into a stable micropeptide using an in vitro transcription/translation assay and overexpression in cells. Linc013026-68AA is predicted to encode a micropeptide of 8 kDa [21]. The in vitro transcription/translation assay with Linc013026-68AA revealed a single band at 8–10 kDa (Figure 3A, arrow). Furthermore, to examine whether Linc013026-68AA protein is stable in cells, we transfected HeLa cells with C-terminal-GFP- and Myc-tagged Linc013026-68AA. GFP-specific immunoblot revealed a band at ~38 kDa for GFP-tagged Linc013026-68AA that corresponds to a molecular mass of 10 kDa for Linc013026-68AA (Figure 3B, 68AA-GFP), while a band of ~15 kDa was observed for Myc-tagged Linc013026-68AA (Figure 3C). Since some proteins can form a dimer in SDS-PAGE (0.1% SDS) [22], we utilized GST pull down assay to examine the interaction of N-terminal GST -tagged 68AA with C-terminal Myc-tagged 68AA. As shown in Figure 1D, no interaction between GST-68AA and 68AA-Myc was detected. In addition, we also observed a ~15 kDa band for 68AA-Myc using cell lysates pre-treated with 1% and 2% SDS under reduced condition (Figure 3E). These data suggested that Myc-tagged Linc013026-68AA did not form a dimer. We then examined the peptide sequence of Linc013026-68AA. Linc013026-68AA contains five potential serine, two threonine and one tyrosine phosphorylation sites (Appendix A) and one lysine acetylation site (G-AcK) [23]. To clarify whether phosphorylation affects the migration of Linc013026-68AA in SDS PAGE, we treated cell lysates with Lambda Protein Phosphatase (Lambda PP) that dephosphorylates phospho-tyrosine, serine and threonine residues. Upon Lambda PP treatment, we observed two additional bands of ~14 and ~10 kDa (Figure 3F (*)), suggesting that phosphorylation contributes to the slower migration of Linc013026-68AA in SDS PAGE.

To examine the endogenous expression of Linc013026-68AA we generated a rabbit antibody against two mixed synthetic peptides corresponding to amino acid positions 4–17 (peptide I) and 54–68 (peptide II) of Linc013026-68AA (Kaneka Eurogentec S.A. Belgium) (amino acid sequences are shown in Figure 3G). First, we tested the specificity of our antibodies. By immunoblot using anti-peptide I and peptide II antibodies, a 38 kDa band for GFP-tagged Linc013026-68AA was specifically detected (Figure 3H). This band was not detected by peptide absorbed antibody (Figure 3H, anti-peptide II + peptide II). We then examined the subcellular localization of exogenous and endogenous Linc013026-68AA using immunofluorescent (IF) and immunohistochemical (IHC) staining. HeLa cells were transfected with Myc-tagged Linc013026-68AA and stained using the immunofluorescent technique with anti-Linc013026-68AA and Myc-specific antibodies. Myc-tagged Linc013026-68AA was detected mainly at the perinuclear region by a Myc-specific staining (Figure 3I). Anti-peptide II but not peptide I antibody gave a strong IF staining signal that completely overlapped with the Myc-specific signal (Figure 3I, Merged). Next, we tested Linc013026-68AA antibodies for IHC staining. In agreement with IF staining, anti-peptide II but not peptide I antibody gave a strong signal for Myc-tagged Linc013026-68AA at the perinuclear region (Figure 3J).

Thus, we used anti-peptide II antibody to examine the endogenous expression of Linc013026-68AA in HepG2 cells. We first depleted Linc013026-68AA using siRNA in HepG2 cells (Figure 3K) and performed anti-peptide II specific immunoblot. In control cells, three major bands ranging from 10–15 kDa were detected (Figure 3L, (*)). Upon Linc013026-68AA depletion, the intensity of these bands was reduced. These data suggested that endogenous Linc013026-68AA may also be phosphorylated as observed for exogenous Linc013026-68AA (Figure 3F). We also tested the endogenous expression of Linc013026-68AA using immunohistochemical staining with anti-peptide II antibody. In control cells, endogenous Linc013026-68AA was detected mainly at the perinuclear region (Figure 3M, siCr) which agreed with the subcellular localization of exogenous Linc013026-68AA. Upon Linc013026-68AA depletion, staining signals of Linc013026-68AA were clearly reduced (Figure 3M, si68AA). In sum, immunoblotting and IHC staining suggested that Linc013026-68AA is endogenously translated into 68AA micropeptide.

### 2.4. Linc013026-68AA Enhances Cell Proliferation

Recent data from our lab suggested that protein derived from lncRNA associates with a biological function [11]. Since HeLa cells do not express Linc013026-68AA (Figure 3I), Myc-tagged Linc013026-68AA in HeLa cells was expressed (Figure 4A). We next examined whether Linc013026-68AA influences cell growth by crystal violet staining assay and Wst-1 assay. Here, within 2 days, growth of Linc013026-68AA-overexpressing HeLa cells was approximately 1.7-fold by crystal violet assay and 1.6-fold by WST assay greater than control vector transfected HeLa cells (Figure 4B,C). Furthermore, depletion of Linc013026 RNA in HepG2 cells reduced cell proliferation approximately two-fold within 3 days measured by crystal violet- (Figure 4D) and Wst-1 assay (Figure 4E), suggesting that Linc013026-68AA promotes cell proliferation. We next examined the Linc013026-68AA transcript in several HCC cell lines, such as HepG2, Hep3B, C3A, Huh7 and HLE. HeLa cells were used as negative control. Linc013026-68AA is expressed in two out of five HCC cell lines (Figure 4F). Thus, we overexpressed Linc013026-68AA in Huh7 and HLE cells, two HCC cell lines that express Linc013026-68AA at low level. These cells also showed an increase in proliferation (1.8-fold in Huh7 cells and 1.6-fold in HLE cells) (Figure 4G), suggesting again that Linc013026-68AA promotes cell proliferation.

## 3. Discussion

In most human cancers, a large number of proteins with driver mutations are involved in tumor development, implying that multiple fine tuners are involved in cancer formation and/or maintenance. A useful strategy for cancer therapy may therefore be to target multiple cancer-specific fine tuners. In this study, using hepatocellular carcinoma as a system we utilized RNA-seq and polysome profiling to identify novel micropeptides derived from cancer-specific lncRNAs. We identified nine lincRNAs that are exclusively expressed in HCC cells but not in normal liver and other tissues (Appendix A). Three out of nine lincRNAs encode small ORFs longer than 50 amino acids and are enriched in polysome fractions (Figure 2C), suggesting that they are translated in a cancer-specific manner. Using a peptide specific antibody we characterized NONHSAT013026.2/Linc013026-68AA, one of our candidates. We show that Linc013026-68AA encodes a 68 amino acid micropeptide that is mainly localized at the perinuclear region (Figure 3I,J,M). Linc013026-68AA is expressed in a subset of HCC cells and plays a role in cell proliferation (Figure 4). We are currently performing interactome analysis of Linc013026-68AA to gain insights into molecular mechanism(s) of Linc013026-68AA. It has been shown that a micropeptide is involved in muscle performance [12] and growth [13]. In addition, SPAR polypeptides encoded by the Linc00961 regulate mTORC1 and muscle regeneration [24], and another micropeptide, mitoregulin, is involved in protein complex assembly in mitochondria [25]. Recently, we demonstrated that C20orf204-189AA encoded by a lincRNA, Linc00176 stabilizes nucleolin and promotes ribosomal RNA transcription [11]. These findings shed light on the role of the previously ignored ‘dark proteome’ in the maintenance of cancer. Thus, further characterization of the coding potency of other cancer-specific lincRNAs (Table 1) may provide clues for identification of novel cancer-specific fine tuners. Furthermore, micropeptides encoded by cancer-specific lncRNAs may also be useful biomarkers for cancer diagnosis.

Why is the expression of a subset of lncRNAs cancer-specific? Recent data identified 161 putative driver genes that are associated with 11 recurrently altered pathways in HCC development [4], and these mutations were not observed in chronic hepatitis or cirrhosis (preneoplastic stages). Interestingly, 28% of the altered gene products play a role in chromatin remodeling, suggesting that abnormal chromatin remodeling results in a cancer-specific expression of a subset of genes [7,8,26]. Indeed, DNase-seq data which map the chromatin accessibility revealed that chromatin at the putative promotor region of seven out of nine HCC-specific lincRNAs is opened in HCC but not in normal liver (Figure 1B). Accessible promoters then enable the recruitment of transcription factors which subsequently activate the transcription in these genes. These data also suggest that cancer cells exhibit remarkable transcriptome alterations, partly by adopting cancer-specific chromatin remodeling events.

One of limitations of this study is the lack of clinical data of HCC-specific lncRNA candidates. Examining the expression of these lncRNAs in RNA-seq data of primary HCC generated by The Cancer Genome Atlas (TCGA) or The International Cancer Genome Consortium (ICGC) will provide clues whether they could be a potentially suitable HCC-specific biomarker. However, retrieving expression from open-access data resource requires the gene annotation by GENCODE [27], while many NONCODE lncRNA genes including lncRNA candidates identified in this study are not yet annotated by GENCODE [28]. Thus it is currently not possible to retrieve expression of our lncRNA candidates from open-access data resource. We are currently examining the protein expression of Linc013026-68AA in primary HCC samples and tumor adjacent normal liver tissues to determine whether it can be a potential HCC biomarker. Furthermore, the role of Linc013026-68AA in in vivo tumor growth should also be examined to clarify whether it may be suitable as a HCC-specific target molecule.

Our study offers novel target molecules as well as biomarkers originating from noncoding RNAs to develop a novel strategy for cancer treatment that targets multiple cancer type-specific fine tuners.

## 4. Materials and Methods

### 4.1. Cell Culture, siRNA, and Transfection

HepG2, Huh7, HLE, C3A and HeLa cells were purchased from the American Type Culture Collection (ATCC, Manassas, VA, USA) or the DMSZ-German collection of microorganisms and cell culture (DMSZ, Braunschweig, Germany). They were grown in DMEM supplemented with 10% FCS. All cell lines are free of mycoplasma contamination.

Control siRNA (5′-UAAGGCUAUGAAGAGAUAC-3′), siLinc013026 (5′-AUGGUGUCAGCAUGUGGAU-3′) were purchased from Microsynth AG (Microsynth AG, Balgach, Switzerland). Fifty picomoles of each siRNA were transfected using Lipofectamin 3000 (Thermo Fisher Scientific, Waltham, MA, USA). For ectopic expression of Linc013026-68AA experiments, Linc013026-68AA cDNA was isolated from HepG2 RNA by RT-PCR. The PCR-product was then cloned into pcDNA3.1 MycHis or pEGFP-N1 vector.

### 4.2. Peptide-Specific Antibodies

Antibodies against the mixture of two synthetic peptides corresponding to amino acid positions 4–17 (peptide I) and 54–68 (peptide II) of Linc013026-68AA were generated in rabbits by Kaneka Eurogentec S.A. (Kaneka Eurogentec S.A., Seraing, Belgium). Two peptide columns were applied for further purification of 4–17 (peptide I) and 54–68 (peptide II) specific antibodies.

### 4.3. Wst-1 Assay

HeLa cells (500–2000 cells/well) were seeded in duplicate on a 96-well plate and then transfected with vector control and Linc013026-68AA and incubated for 2 days. A Wst-1 proliferation assay kit (Roche Diagnostics, Basel, Switzerland) was employed according to the manufacturer’s instructions.

### 4.4. Crystal Violet Assay

HeLa, HepG2, Huh7 and HLE cells (500–2000 cells/well) were seeded in duplicate on a 96-well plate and then transfected with vector control, Linc013026-68AA or siRNAs and incubated for 2 days. Cells were then washed with phosphate-buffered saline (PBS) and fixated with methanol. Crystal violet dye was applied for 10 min. After air drying the plate, the dye was solubilized in methanol and absorbance was measured at 595 nm.

### 4.5. Immunohistochemistry/Immunofluorescence

Immunohistochemical and immunofluorescent studies were performed as detailed previously [5,29]. Rabbit monoclonal anti-Myc antibody was from Cell Signaling Technology (cs-2278S, Cambridge, UK).

### 4.6. In Vitro Transcription/Translation

Radiolabeled substrates were generated by in vitro transcription/translation using the plasmid pcDNA3.1-Linc013026-MYC, the SP6/T7-coupled TNT reticulocyte lysate system (Promega, Madison, WI, USA), and [^35^S]methionine (370 kBq/μL, >37 TBq/mmol, Hartmann Analytic, Braunschweig, Germany) according to the manufacturer’s instructions.

### 4.7. mRNA Export Assay

Isolation of nuclear- and cytoplasmic RNA was performed as previously described [30,31]. Briefly, cells were washed with ice-cold PBS three times and incubated in cytoplasmic buffer (100 mm Tris-HCl pH 8.0, 150 mm NaCl, 0.5% (*v*/*v*) NP-40, protease inhibitor cocktail [Sigma-Aldrich, St. Louis, MO, USA]) and RNase inhibitor (NEB, Ipswich, MA, USA) for 5 min on ice. Cells were then harvested. Nucleus were pelleted by centrifugation. Nuclear- and cytoplasmic RNAs were isolated using the ReliaPrep^TM^ miRNA cell and tissue miniprep system (Promega, Madison, WI, USA) according to the manufacturer’s instructions.

### 4.8. Polysome Profiling

Polysome fractions were prepared using sucrose gradient fractionation as previously described [32]. To prepare polysomes, 1.25 × 10^7^ HepG2 cells were rinsed and scraped in ice-cold PBS containing cycloheximide (0.1 mg/mL). Subsequent steps were carried out in the cold. After pelleting by centrifugation at 500× *g* for 7 min, the cells were resuspended in extraction buffer (20 mm Tris-HCl, pH 8.0, 140 mm KCl, 0.5 mm DTT, 5 mm MgCl_2_, 0.5% Nonidet-P40, 0.1 mg/mL cycloheximide, and 0.5 mg/mL heparin) and incubated for 5 min on ice. Extracts were centrifuged for 10 min at 12,000× *g*. Approximately 0.5 mL of supernatant was layered onto a 12-mL linear sucrose gradient (10–50% sucrose (*w*/*v*) in 20 mm Tris-HCl, pH 8.0, 140 mm KCl, 0.5 mm DTT, 5 mm MgCl_2_, 0.1 mg/mL cycloheximide, and 0.5 mg/mL heparin) and centrifuged at 4 °C in an SW40Ti rotor (Beckman, Palo Alto, CA, USA) at 35,000 rpm without brake for 80 min (120 min for experiments examining the distribution of β-globin reporter mRNAs). The gradients were collected into 10–12 1-mL fractions, and absorbance profiles at 260 nm were recorded (ISCO, UA-6 detector). An amount of 0.1 volume of 3 m sodium acetate (pH 5.2) and 1 volume of isopropyl alcohol were added to the probes for overnight precipitation at −20 °C. RNA was purified using the ReliaPrepTM miRNA cell and tissue miniprep system (Promega, Madison, WI, USA).

### 4.9. Immunoblotting Procedures

Details of immunoblotting have been described previously [31]. Corresponding proteins were visualized by incubation with peroxidase-conjugated anti-mouse, anti-rabbit or anti-goat immunoglobulin, followed by incubation with SuperSignal West FemtoMaximum Sensitivity Substrate (Thermo Fisher Scientific, Waltham, MA, USA). Results were documented on a LAS4000 imaging system (GE Healthcare BioSciences, Uppsala, Sweden). Mouse monoclonal anti-Myc (9E10), anti-GAPDH (sc-32233), anti-GFP (sc-9996) and polyclonal anti-Actin (sc-1616) were purchased from Santa Cruz Biotechnology (Santa Cruz, CA, USA). Polyclonal anti-Histone H3 was from Cell Signaling.

### 4.10. Semi-Quantitative RT-PCR and qRT-PCR Analysis

RNA was isolated from cells with the ReliaPrep^TM^ miRNA cell and tissue miniprep system (Promega, Madison, WI, USA) according to the manufacturer’s instructions. One microgram of RNA was reverse-transcribed using oligo dT primer or random primer and the ProtoScript^®^ II Reverse Transcriptase (NEB, Ipswich, MA, USA) following the instructions provided. One-twentieth of the cDNA mix was used for real-time PCR with 10 pmol of forward and reverse primer and ORA^TM^ qPCR Green Rox kit (HighQu, Kraichtal, Germany) in a Qiagen Rotorgene machine. The levels of mRNA expression were standardized to the glyceraldehyde-3 phosphate dehydrogenase (GAPDH) mRNA level. Primer sequences are shown in Appendix A.

### 4.11. Statistical Analysis

Cell experiments were performed in triplicate and a minimum of three independent experiments were evaluated. Data were reported as the mean value with standard deviation. The statistical significance of the difference between groups was determined by Student’s t-test (two sided).

### 4.12. RNA Sequencing Data Analysis

Raw sequencing data (FASTQ files) were downloaded from the ENCODE portal or Gene Expression Omnibus (GEO). Galaxy workflow for RNA-Seq (www.usegalaxy.org) (accessed on 20 November 2020) was used for subsequent data analysis. Reads were mapped to the human reference genome (GRCh38) using Bowtie2 (Galaxy Version 2.3.4.1). The gene expression values (Fragments per Kilobase Million (FPKM)) were calculated by Cuffnorm (Galaxy Version 2.2.1.5) using the human NONCODEv5 transcript reference.

### 4.13. GST Pull-Down Assay

HeLa cells were transfected with pcDNA3.1-Linc013026-MYC for one day and lysed with lysis buffer (10 mm Tris, 150 mm NaCl, 1 mm PMSF, 0.4% NP40, protease inhibitor cocktail (Sigma-Aldrich, Munich, Germany). After centrifugation, supernatants were incubated with GST and GST-Linc013026-68AA fusion protein. Bound proteins were analyzed by Myc- and GST-specific immunoblot.

## Figures and Tables

**Figure 1 ijms-23-00058-f001:**
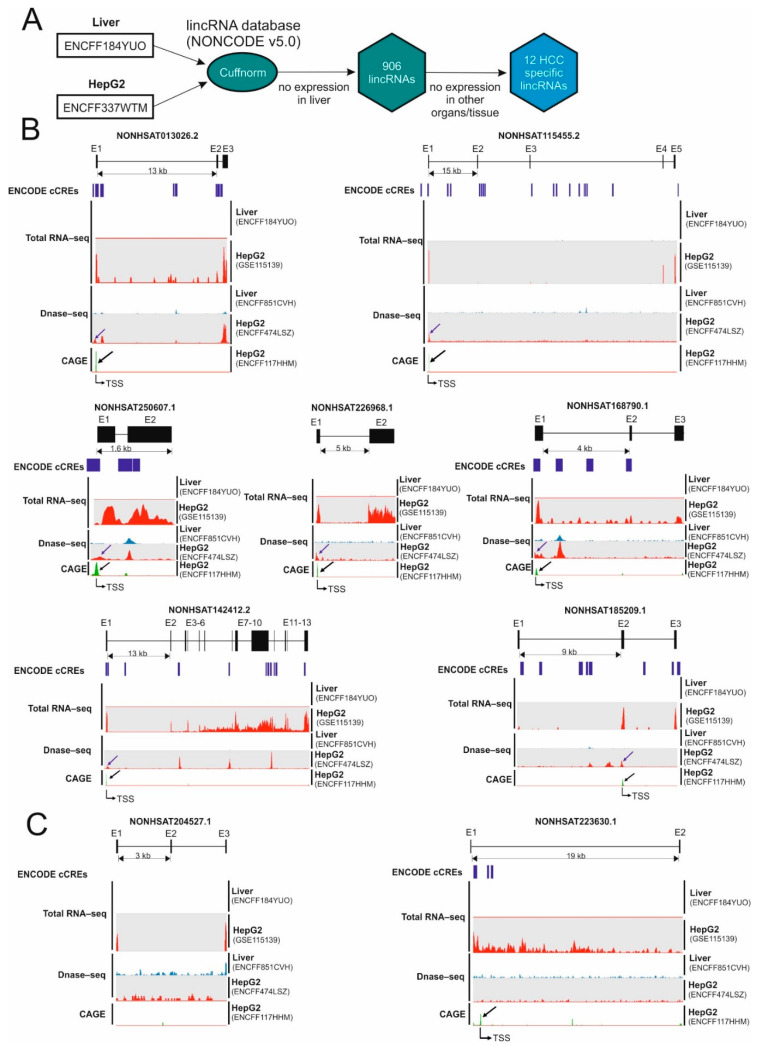
Identification of hepatocellular carcinoma-specific lncRNAs. (**A**) RNA-sequencing (RNA-seq) data from normal liver (ENCFF184YUO) and from the HCC cell line HepG2 (ENCFF337WTM) were aligned to the human reference genome (GRCh38) using Bowtie2. The gene expression values (Fragments per Kilobase Million (FPKM)) were calculated by Cuffnorm using the human NONCODE v5.0 transcript reference. A total of 906 long intergenic noncoding RNAs (lincRNAs) that were expressed in HepG2 cells but not in the normal liver were selected. Among these 906 lincRNAs twelve lincRNAs were expressed exclusively in HepG2 cells but not in the liver and other organs/tissues. (**B**,**C**) Abnormal chromatin structure induced expression of HCC-specific lincRNAs: Total RNA-seq from liver (ENCFF184YUO) and HepG2 cells (GSE115139), DNase-sequencing (DNase-seq) from normal liver (ENCFF851CVH) and HepG2 (ENCFF474LSZ) and candidate cis-Regulatory Elements (cCREs) generated by ENCODE consortium (ENCODE cCREs, blue mark) and cap analysis of gene expression (CAGE) data in HepG2 cells (ENCFF177HHM) were aligned to the human reference genome (GRCh38). SeqMonk was used to quantitate and visualize the data. Peaks in the wiggle plot represent the normalized RNA-seq, DNase-seq and CAGE read coverage on HCC-specific lincRNAs. E: exon; blue arrow: Open chromatin region at the putative promoter; black arrow: transcription start site (TSS) detected by CAGE.

**Figure 2 ijms-23-00058-f002:**
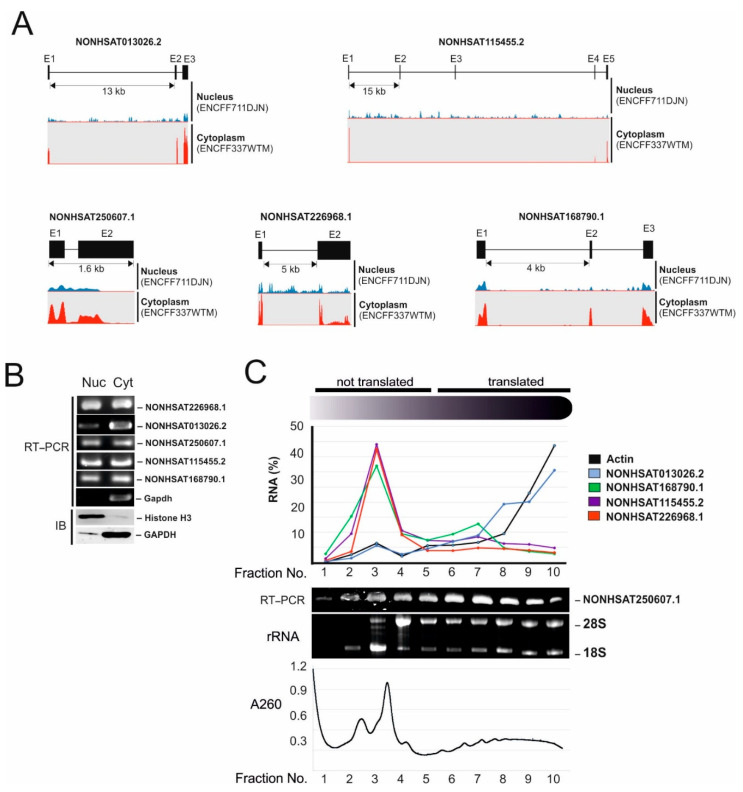
Identification of micropeptide candidates derived from HCC-specific lincRNAs. (**A**) Nuclear- (ENCFF711DJN) and cytoplasmic (ENCFF337WTM) RNA-seq of HepG2 cells generated by the ENCODE Consortium were aligned to the reference human genome (GRCh38). SeqMonk was used to quantitate and visualize the data. Peaks in the wiggle plot represent the normalized RNA-seq read coverage on HCC-specific lincRNAs. E: exon. (**B**) RNA was isolated from the nuclear (Nuc) and cytoplasmic (Cyt) fractions of HepG2 cells and analyzed by RT-PCR. Fractionation quality was measured by immunoblot analysis of THOC5, GAPDH and Histone H3 (Blot). Three independent experiments were performed. (**C**) HepG2 cytoplasmic lysate was prepared and fractionated on sucrose gradients. The distribution of RNA was calculated using the CT values obtained by qRT-PCR. Isolated RNA was supplied in a gel to determine translated fractions. mRNAs were prepared from the indicated fractions and were applied for Actin, NONHSAT013026.2, NONHSAT168790.1, NONHSAT115455.2, and NONHSAT226968.1 qRT-PCR or NONHSAT250607.1 RT-PCR. A representative absorbance profile at 260 nm was obtained during fractionation of gradients. A replicate is shown in Appendix A.

**Figure 3 ijms-23-00058-f003:**
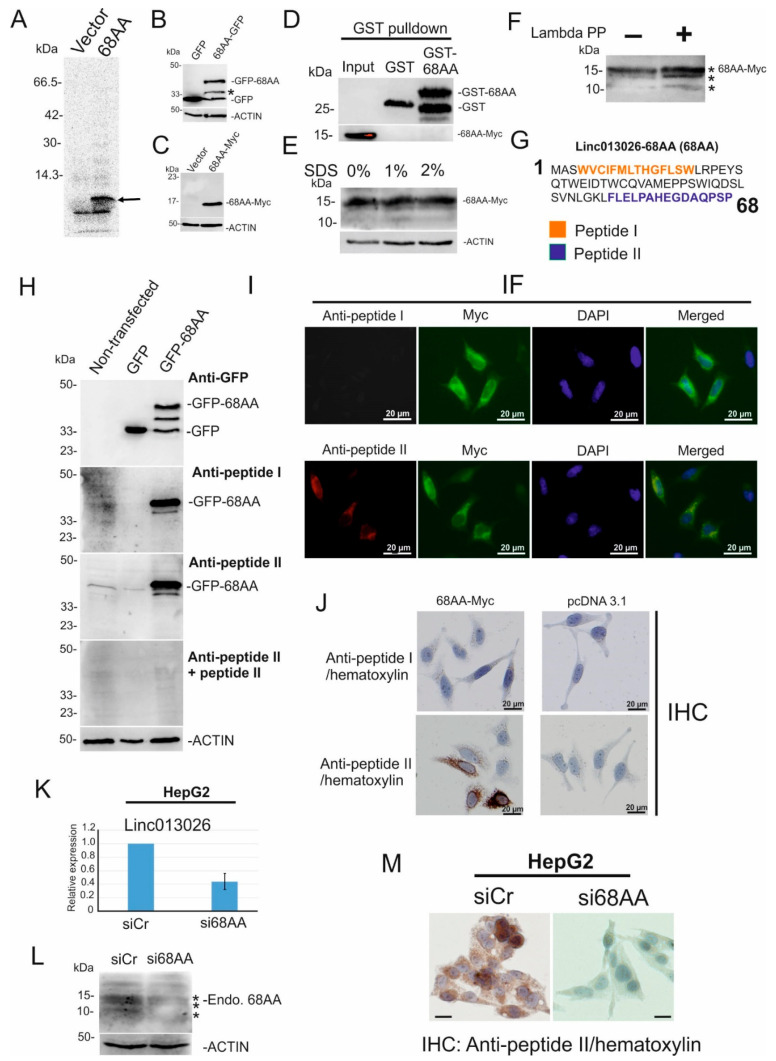
NONHSAT013026.2/Linc013026-68AA is translated into a 68 amino acid long micropeptide. (**A**) In vitro transcription/translation assay of Linc013026-68AA, proteins are labeled with [^35^S]methionine. Arrow indicated the translated peptide. (**B**,**C**) GFP- and Myc-tagged Linc013026-68AA (68AA) and pEGFP-N1 (GFP) or pcDNA3.1 MycHis (Vector) vector was transfected in HeLa cells, and GFP- and Myc-specific immunoblot were performed. (**D**) 68AA-Myc was overexpressed in HeLa cells. Cell extracts were incubated with GST and GST-68AA and then Myc and GST specific immunoblots were performed. (**E**) 68AA-Myc overexpressing cell extracts were pre-treated with 1% and 2% SDS and then Myc and ACTIN specific immunoblots were performed. (**F**) 68AA-Myc was overexpressed in HeLa cells. Forty microliters of cell extracts were treated with 2000 U Lambda Protein Phosphatase (Lambda PP) for 45 min at 30 °C and subsequently supplied for Linc013026-68AA specific immunoblot. (**G**) Amino acid sequence of Linc013026-68AA was depicted. Numbers represent amino acid number. Amino acid sequences of peptide I (orange) and peptide II (blue) were used to generate rabbit antibodies. (**H**) Non-transfected-, pEGFP-N1 (GFP) and GFP-tagged Linc013026-68AA (68AA-GFP) HeLa cell lysates were applied for GFP-, anti-peptide I and anti-peptide II or anti-peptide II + peptide II immunoblot. ACTIN was used as loading control. (**I**) Myc-tagged Linc013026-68AA was expressed in HeLa and stained with anti-Myc, anti-peptide I and anti-peptide II specific antibodies and visualized by the immunofluorescent (IF) technique. (**J**) Myc-tagged Linc013026-68AA (68AA-Myc) and pcDNA3.1 MycHis (Vector) vector were transfected in HeLa cells and immunohistochemically stained with rabbit antibody against Linc013026-68AA-peptide I and peptide II and hematoxylin. (**K**) HepG2 cells were transfected with siCr and siLinc013026-68AA for three days. Total RNAs were isolated and supplied for Linc013026 or Gapdh-specific qRT-PCR. The expression of Linc013026 was normalized by Gapdh. Three independent experiments were performed. Numbers are mean ± standard deviation (SD). (**L**) A sister culture of (**K**) was subjected to anti-peptide II and ACTIN specific immunoblot. (**M**) A sister culture of (**K**) was immunohistochemically stained with anti-peptide II and hematoxylin. All bars represent 20 µm. (*) changes of LINC013026-68AA migration in SDS-PAGE induced by phosphorylation.

**Figure 4 ijms-23-00058-f004:**
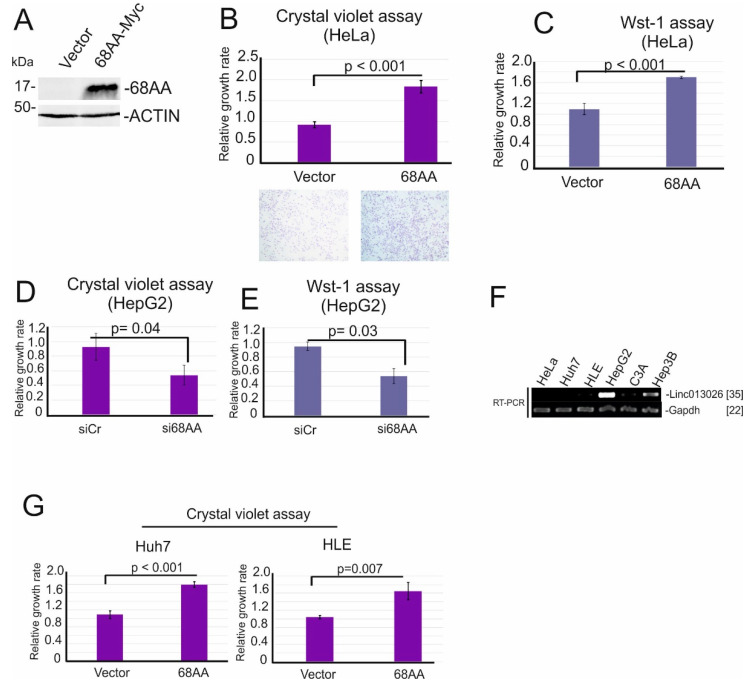
Linc013026-68AA enhances cell proliferation. (**A**) HeLa cells were transfected with Myc-tagged Linc013026-68AA (68AA-Myc) and pcDNA3.1 MycHis (Vector) vector for two days and anti-Linc013026-68AA- and Actin-specific immunoblot were performed. ACTIN was used as loading control. (**B**,**C**) Sister cultures of (**A**) were supplied for crystal violet- (**B**) and Wst-1 assay (**C**) to examine the effect of Linc013026-68AA on cell proliferation. (**D**,**E**) HepG2 cells were transfected with siRNA control (siCr) and siLinc013026 (si68AA) for three days and supplied for crystal violet- (**D**) and Wst-1 assay (**E**). (**F**) Total RNAs were isolated from HeLa, Huh7, HLE, HepG2, C3A and Hep3B cells. The expression of Linc013026 was examined using RT-PCR. Gapdh mRNA was used as loading control. [] indicates number of PCR cycles. (**G**) Huh7- and HepG2 cells were transfected with Myc-tagged Linc013026-68AA (68AA-Myc) and pcDNA3.1 MycHis (Vector) vector for two days and supplied for crystal violet- and Wst-1 assay. Three independent experiments were performed for crystal violet- and Wst-1 assay. Numbers are mean ± standard deviation (SD). p: *p*-value.

**Table 1 ijms-23-00058-t001:** List of six HCC-specific lincRNAs that contain at least one ORF longer than 50AA.

Transcript ID	Expression in the Liver (FPKM) (ENCFF184YUO)	Expression in HepG2 (FPKM) (ENCFF337WTM)	Number of ORFs Longer Than 50AA
NONHSAT226968.1	0	104.9	2
NONHSAT013026.2/Linc013026	0	61.3	2
NONHSAT250607.1	0	39.3	2
NONHSAT142412.2	0	30.5	12
NONHSAT115455.2	0	15.0	2
NONHSAT168790.1	0	13.7	3

## Data Availability

All data generated or analyzed during this study are included in this published article and its additional files.

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
