# Peer review of "Identification of Novel Micropeptides Derived from Hepatocellular Carcinoma-Specific Long Noncoding RNA"

_ijms, 2021, doi:10.3390/ijms23010058_

Round 1

Reviewer 1 Report

This study utilized RNA sequencing (RNA-seq) and polysome profiling to identify a novel 68AA long micropeptide encoded by Linc013026. The author confirmed the expression of  Linc013026-68AA by raidoisotope-based in vitro translation assay and IHC staining in human cells. Furthermore, they showed the role of the 68AA on growth of cancer cells by RNAi or overexpression according to the basal level of various cell types. The design is delicate and the manuscript is well-written.  Here I have some questions.

Minor points

  1. Could you provide clinical value of Linc013026 or 68AA retrieved from open-access data resource, e.g. TCGA-LIHC? This study would even more appealing if clinical data could be revealed. 
  2. Figure 4F: What do the "[35]" and "[22]" refer to?
  3. Have you look into what mechanisms account for the growth-promoting effect of 68AA in HCC cell lines?

Reviewer 2 Report

The Authors have conducted a very complex study using RNA-sequencing and polysome profiling to identify novel micropeptides that originate 19 from HCC-specific lncRNAs. The results are sound.

However, I would suggest the Authors to greatly expand on the potential relevance and implications for clinical practice - the reader is left with some feeling of "so what?" after reading the paper.

A very minor comment, paragraph 4.11 is entitled "Statistical analysis and limitation of the study", but no limitation is commented in the paragraph. I suggest the authors to better discuss the limitations in the Discussion.

Reviewer 3 Report

In this manuscript, the authors identified a lncRNA that is expressed exclusively in HepG2 cells. Although it is very interesting, the data are preliminary, and further experiments are necessary to draw a conclusion. The following concerns might be helpful to improve this study.

1. lines 157-158: I think that SDS-PAGE was performed in Figure 3C. How did the authors conceive that Myc-tagged protein formed a dimer? What about other post-translational modifications?

2. Western blotting for endogenous 68AA should be examined.

3. Figure 1: The authors should search for consensus sequences among the putative promoter regions.

The authors should investigate the mechanism that control Linc013028 in HepG2 cells. Did the RNA seq data suggest such mechanisms?

Round 2

Reviewer 1 Report

The authors' responses are appreciated. I recommend acceptance for publication in present form.

Reviewer 3 Report

The authors have sufficiently addressed my concerns.